# Estimating personal temporal symptom networks for childhood cancer survivors

Yiwang Zhou [1] ✉, Samira Deshpande[1], Madeline R. Horan[2], Jaesung Choi[3], Daniel A. Mulrooney [3,4], Kirsten K. Ness [3], Melissa M. Hudson[4], Deo Kumar Srivastava[1,5] & I-Chan Huang [3,5] ✉

## Abstract

**Background** Childhood cancer survivors experience persistent and evolving symptom burden post-therapy. Network analysis can help uncover the complex symptom patterns. However, current network analyses often rely on cross-sectional data and focus on average symptom patterns among survivors, overlooking individual heterogeneities.

**Methods** We introduced an autoregressive logistic model with covariates to account for individual heterogeneities in network estimation and to construct personal temporal symptom networks. Simulation experiments were conducted to validate the robustness of this method in constructing personal temporal symptom networks. We also applied the autoregressive logistic model with covariates to longitudinal symptom data from a random sample of 2000 adult survivors of childhood cancer in the St. Jude Lifetime Cohort Study (SJLIFE).

**Results** Simulation studies demonstrate that the proposed method reliably recovers personal temporal symptom network structures under various conditions. In the real data application, older age, female sex, lower educational attainment, annual personal income < $20,000, and receipt of chemotherapy and/or radiation therapy are associated with stronger connections between symptoms at baseline and the first follow-up.

**Conclusions** We demonstrate that the logistic autoregressive model with covariates effectively estimates personal temporal symptom networks for childhood cancer survivors, enabling personalized symptom monitoring and informing tailored symptom management strategies.

## Plain Language Summary

Childhood cancer survivors often face ongoing physical and psychological symptoms that can persist or change over time. In this study, we developed a statistical method to better understand how these symptoms are linked and how these connections evolve over time for each survivor. This study tested the method using computer simulations and applied it to real data from 2000 adult survivors of childhood cancer. We found that personal factors, such as age, sex, education, income, and treatment history, can influence how symptoms are connected over time. These findings show the importance of considering each survivor's unique symptom experience and help doctors create more personalized strategies to monitor and manage long-term symptom burden for cancer survivors.

Children with cancer frequently report substantial physical and psychological symptoms during therapy. These symptoms may persist after therapy completion and new symptoms may emerge[1,2]. The prevalence of multiple symptoms reported by childhood cancer survivors is approximately 87%[2] and may be interrelated, forming complex systems with distinct structures and patterns, known as symptom networks[3–5]. Instead of viewing individual symptoms in isolation as indications of adverse events (e.g., metastasis)[6–8], examining patterns of interconnected symptoms may enhance our understanding of how co-existing symptoms precede clinical identification of late effects[9,10]. Identifying the central symptoms within symptom networks will facilitate interventions targeted at specific late effects for effective disease management[10].

Conventional symptom research and clinical applications focus on evaluating individual symptoms (e.g., pain, fatigue, and sleep disturbance). However, this approach cannot directly infer underlying interconnections between multiple symptoms that can be estimated through a network approach with data from a patient cohort. While various methods have been proposed for symptom network analysis, previous research has often been performed using cross-sectional symptom data, resulting in networks that depict interconnections between symptoms at a single point in time[11–15]. As the volume of biomedical and clinical data available through electronic health records continues to grow, time-series data are increasingly accessible to researchers and clinicians. These longitudinal symptom data enable a deeper understanding of the dynamic evolution of symptom networks over the disease course.

[1]Department of Biostatistics, St. Jude Children's Research Hospital, Memphis, TN, USA. [2]Department of Pediatrics, School of Medicine, Wake Forest University, Winston-Salem, NC, USA. [3]Department of Epidemiology and Cancer Control, St. Jude Children's Research Hospital, Memphis, TN, USA. [4]Department of Oncology, St. Jude Children's Research Hospital, Memphis, TN, USA. [5]These authors contributed equally: Deo Kumar Srivastava, I-Chan Huang. ✉e-mail: yiwang.zhou@stjude.org; i-chan.huang@stjude.org

Utilizing time-series data, Epskamp 2020[16] presents a general framework for constructing three distinct network types, including the temporal, contemporaneous, and between-subject network. Through the graphical vector-autoregression model[17,18], temporal relationships among symptom experiences can be modeled via regression on preceding measurement occasions. This approach yields a regression matrix that enables the creation of a network with directed edges, known as a *temporal network*. This temporal network illustrates how each symptom predicts the recurrence of its symptom or the occurrence of other symptoms over time. The remaining variances and covariances in symptom data can be modeled as a Gaussian graphical model (GGM)[17,19], known as the *contemporaneous network*. In scenarios where time-series symptom data from multiple subjects are available, another GGM can be formed to capture the between-subject effects, known as the *between-subject network*. In this paper, we focus on analyzing the *temporal network* of symptoms experienced by adult survivors of childhood cancer.

Although the fundamentals of estimating temporal network using time-series data have been thoroughly investigated in Epskamp 2020[16], the construction of temporal network was based on data from all individuals without accounting for individual heterogeneities that could influence the co-occurrence and magnitude of different symptoms within the estimated network[20–22]. Extending from our recent work that incorporated personal, clinical, socio-demographic and neighborhood risk factors into the construction of a cross-sectional symptom network[5], the *first* objective of this study was to model the interconnections between symptoms within the temporal network while accounting for individual differences using the logistic autoregressive model with covariates. We conducted extensive simulation experiments to comprehensively evaluate the effectiveness of this modeling approach and bootstrap testing in inferring the temporal symptom networks.

The *second* objective of this study was to estimate personal temporal symptom network using longitudinal symptom data collected from adult survivors of childhood cancer enrolled in the St. Jude Lifetime Cohort Study (SJLIFE)[23,24]. Our goal was to create a personal temporal network for each survivor based on the evolution of the survivor's symptoms over time across ten domains: cardiac symptoms, pulmonary symptoms, sensation abnormality, nausea, movement problems, pain, memory problems, fatigue, anxiety, and depression. By integrating individual socio-demographic characteristics and treatment information, we estimated the covariate-related pairwise associations between symptoms over time. Our personal temporal symptom network modeling approach allows us to discern the personal factors influencing symptom associations over time and to pinpoint sentinel symptoms that substantially shape the overall temporal network structure among childhood cancer survivors.

Our final results demonstrate that logistic autoregressive model with covariates can effectively estimate personal temporal symptom networks for childhood cancer survivors, enabling more personalized symptom monitoring and management. Simulation studies confirm that the proposed method reliably recovers individual symptom network structures under various conditions. Applying this method to real symptom data for childhood cancer survivors in the SJLIFE cohort, we find that factors such as older age, female sex, lower educational attainment, lower income, and a history of chemotherapy and/or radiation therapy are associated with stronger symptom connections over time. These findings highlight the importance of considering personal factors when assessing symptom burden and suggest that individualized strategies may improve long-term symptom management in cancer survivors.

## Methods
### Notations
Network analysis, grounded in graph theory, examines a system of interconnections between multiple objects[25,26]. A network consists of two

fundamental components: nodes and edges. In this study, nodes represent binary indicators of symptoms (presence or absence). Edges in a network depict associations between nodes, either undirected or directed. This study focuses on temporal networks with directed edges, indicating the direction of temporal association between nodes. Edges of a network can be weighted, representing the strength of dependency between nodes.

Suppose we have collected symptom data for $n$ subjects, $i = 1, \ldots, n$. For each subject $i$, we obtain symptom vectors $(\boldsymbol{y}_i^1, \boldsymbol{y}_i^2, \ldots, \boldsymbol{y}_i^T)$ at timepoints $t = 1, \ldots, T$. Each symptom vector $\boldsymbol{y}_i^t$ consists of observations of $p$ symptoms, $j = 1, \ldots, p$, with $\boldsymbol{y}_i^t = \left( y_{i,1}^t, y_{i,2}^t, \ldots, y_{i,p}^t \right) \in \{0, 1\}^p$. Here, $y_{i,j}^t = 1$ indicates the presence of a symptom while $y_{i,j}^t = 0$ indicates its absence. Let $\boldsymbol{X_i} = (X_{i1}, \ldots, X_{id})$ represent a $d$-dimensional vector of individual characteristics for subject $i$.

### Temporal symptom network estimation
Utilizing longitudinal binary symptom data, the temporal symptom network can be constructed to illustrate dynamic changes in the network structure over time. This construction can be achieved using the logistic autoregressive model[27]. The probability of a symptom taking the value of 1 at timepoint $t$ conditional on the previous $K$ symptom vectors equals

$$
\begin{aligned}
p\left( y_j^t = 1 | \boldsymbol{y}^{t-1}, \ldots, \boldsymbol{y}^{t-K} \right) &= \text{logit}^{-1} \left( \sum_{k=1}^{K} \sum_{j'=1}^{p} \sigma_{jj'}^k y_{j'}^{t-k} + \mu_j \right) \\
&= \frac{\exp(\sum_{k=1}^{K} \sum_{j'=1}^{p} \sigma_{jj'}^k y_{j'}^{t-k} + \mu_j)}{1 + \exp(\sum_{k=1}^{K} \sum_{j'=1}^{p} \sigma_{jj'}^k y_{j'}^{t-k} + \mu_j)}
\end{aligned}
\tag{1}
$$

Coefficient $\mu_j$ is the threshold that measures the probability of variable $y_j^t$ taking the value of 1 when all the neighbors being zero, and $\sigma_{jj'}^k$ represents the pairwise association between $y_{j'}^{t-k}$ and $y_j^t$ conditional on all the other variables. These elements are effectively logistic regression coefficients linking previous values $(\boldsymbol{y}^{t-1}, \ldots, \boldsymbol{y}^{t-K})$ of the dynamic process to the current values $\boldsymbol{y}^t$. The model is termed as logistic autoregressive model because it is a vector autoregressive model with a logistic link[27]. The parameters of the logistic autoregressive model are the $K$ matrices $\boldsymbol{\Sigma}^1, \ldots, \boldsymbol{\Sigma}^K$ and the vector $\boldsymbol{\mu}$. The log-likelihood of the model is,

$$
\begin{aligned}
l(\boldsymbol{y}|\boldsymbol{\mu}, \boldsymbol{\Sigma}^1, \ldots, \boldsymbol{\Sigma}^K) = \sum_{t=1}^{T} \sum_{j=1}^{p} & \left[ y_j^t \left( \sum_{k=1}^{K} \sum_{j'=1}^{p} \sigma_{jj'}^k y_{j'}^{t-k} + \mu_j \right) \right. \\
& \left. - \log \left( 1 + \exp \left( \sum_{k=1}^{K} \sum_{j'=1}^{p} \sigma_{jj'}^k y_{j'}^{t-k} + \mu_j \right) \right) \right] = \sum_{j=1}^{p} l(\boldsymbol{y}_j|\boldsymbol{\sigma}_{j\cdot}^1, \ldots, \boldsymbol{\sigma}_{j\cdot}^K, \mu_j)
\end{aligned}
\tag{2}
$$

where $l(\boldsymbol{y}_j|\boldsymbol{\sigma}_{j\cdot}^1, \ldots, \boldsymbol{\sigma}_{j\cdot}^K, \mu_j)$ are likelihoods for the parameters associated with the variable $\boldsymbol{y}_j$ to its neighbors and $\boldsymbol{\sigma}_{j\cdot}^k = (\sigma_{j1}^k, \ldots, \sigma_{jp}^k)$. Estimation of the parameters $\boldsymbol{\Sigma}^1, \ldots, \boldsymbol{\Sigma}^K$ and $\boldsymbol{\mu}$ can be made via a total of $p$ logistic regression models, each of which treats a variable $y_j^t$ as the outcome and the status of all variables, including $y_j$ itself, at the previous $K$ timepoints (i.e., $y_{j'}^{t-1}, \ldots, y_{j'}^{t-K}, j' = 1, \ldots, p$) as the predictors. Note that $\boldsymbol{\Sigma}^1, \ldots, \boldsymbol{\Sigma}^K$ are usually asymmetric, resulting in directed networks that can illustrate the temporal direction of prediction. To simplify the modeling of network structure in our study, we set $K = 1$, meaning we only use the symptom data collected at the previous timepoint $t - 1$ to predict the symptom presentation at time $t$. This lag-1 factorization of the logistic autoregressive model results in the parameters being only $\boldsymbol{\mu}$ and $\boldsymbol{\Sigma}^1$.

To further balance the complexity of the temporal network (i.e., the number of parameters to be estimated) with the information available from data, we performed eLASSO by leveraging LASSO[28] to shrink negligible effects to zero, creating a sparse network. The eLASSO procedure determines the optimal structure of a network by minimizing an extended Bayesian Information Criterion (eBIC)[29–31],

expressed as

$$\text{eBIC} = -2l + |J| \log N + 2\eta |J| \log(p) \quad (3)$$

where $l$ is the log-likelihood, $|J|$ denotes the number of neighbors selected by LASSO at a specific tuning parameter $\eta$, $N = n(T-1)$ signifies the number of observations, and $p$ is the total number of covariates in the regression model.

### Personal temporal symptom network estimation

Here we propose the algorithm for estimating personal temporal symptom networks by incorporating covariates into the logistic autoregressive model. Personal temporal symptom network can be estimated by incorporating $X_i$ into the logistic autoregressive model, with the likelihood given by

$$
l(y|\mu, \alpha, \Sigma^1, \gamma^1)
$$
$$
= \sum_{t=1}^{T} \sum_{j=1}^{p} \left[ y_j^t \left( \sum_{j'=1}^{p} \left( \sigma_{jj'}^1 + \sum_{l=1}^{d} \gamma_{jj'l}^1 x_l \right) y_{j'}^{t-1} + \mu_j + \sum_{l=1}^{d} \alpha_{jl} x_l \right) \right.
$$
$$
\left. - \log \left( 1 + \exp \left( \sum_{j'=1}^{p} \left( \sigma_{jj'}^1 + \sum_{l=1}^{d} \gamma_{jj'l}^1 x_l \right) y_{j'}^{t-1} + \mu_j + \sum_{l=1}^{d} \alpha_{jl} x_l \right) \right) \right]
$$
$$(4)$$

The threshold (i.e., probability of a node $y_j^t$ taking value 1 when all other nodes $y_{j'}^{t-1}$ being 0) for variable $y_j^t$ is $\mu_j + \sum_{l=1}^{d} \alpha_{jl} x_l$, and the pairwise association between variables $y_j^t$ and $y_{j'}^{t-1}$ is $\sigma_{jj'}^1 + \sum_{l=1}^{d} \gamma_{jj'l}^1 x_l$, both of which integrate a linear sum of covariates $x_l$. Estimation of parameters $\mu, \Sigma^1, \alpha$ and $\gamma^1$ can be achieved by fitting a total of $p$ logistic regression models, each treating $y_j^t$ as the outcome and status of all variables at the previous timepoint (i.e., $y_{j'}^{t-1}, j' = 1, \ldots, p$) as the predictors. Coefficient $\mu_j$ represents the threshold of $y_j^t$ when all covariates $x_l = 0$, while coefficient $\alpha_{jl}$ signifies the difference in the threshold when $x_l$ changes from 0 to 1. Through $\alpha_{jl}$, we can assess the impact of $x_l$ on the threshold of $y_j^t$. Similarly, coefficient $\sigma_{jj'}^1$ denotes the pairwise association between $y_j^t$ and $y_{j'}^{t-1}$ when all covariates $x_l = 0$, while coefficient $\gamma_{jj'l}^1$ captures the difference of pairwise association between $y_j^t$ and $y_{j'}^{t-1}$ when $x_l$ changes from 0 to 1. Based on $\gamma_{jj'l}^1$, we can evaluate the influence of $x_l$ on the conditional dependency between $y_j^t$ and $y_{j'}^{t-1}$. Utilizing the estimates $\hat{\Sigma}^1$ and $\hat{\gamma}^1$, we can construct personal temporal symptom networks for subjects with heterogeneous characteristics. Similarly, network sparsity can be achieved using the eLasso procedure[29–31] with a tuning parameter $\eta$ that balances the complexity of the temporal network (i.e., the number of parameters to be estimated) with the available data. A detailed algorithm of the proposed logistic autoregressive model with covariates are provided as follows:

**Algorithm 1**. Algorithm of logistic autoregressive model with covariates.

1. Denote a dataset as $D = \left\{ \left( x_{i1}, \ldots, x_{id}, y_{i,1}^1, \ldots, y_{i,p}^1, \ldots, y_{i,1}^T, \ldots, y_{i,p}^T \right), i = 1, \ldots, n \right\}$.
2. For $j = 1, \ldots, p$,
   a. Treated $y_j^t$ as outcome; $y_{j'}^{t-1}, j' = 1, \ldots, p$ and $x_l, l = 1, \ldots d$ as predictors.
   b. Fit a $l_1$-regularized logistic regression with varying penalty parameter $\eta$:

$$
\text{logit} P\left(y_j^t = 1\right) = \mu_j + \sum_{l=1}^{d} \alpha_{jl} x_l + \sum_{j'=1}^{p} \left( \sigma_{jj'}^1 + \sum_{l=1}^{d} \gamma_{jj'l}^1 x_l \right) y_{j'}^{t-1}
$$
$$(5)$$

c. Computed the eBIC value for each $\eta$.
d. Identified $\eta$ that yields the lowest eBIC.
e. Collected the resulting regression parameters $\hat{\mu}, \hat{\alpha}, \hat{\Sigma}^1$, and $\hat{\gamma}^1$.
3. The estimated threshold for variable $y_j$ is $\hat{\mu}_j + \sum_{l=1}^{d} \hat{\alpha}_{jl} x_l$. The estimated association for $y_{j'}^{t-1}$ in predicting $y_j^t$ is $\hat{\sigma}_{jj'}^1 + \sum_{l=1}^{d} \hat{\gamma}_{jj'l}^1 x_l$. Note that when $j = j'$, the association is the estimated self-sustained probability.

### Evaluation of network accuracy and stability

After constructing a personal temporal symptom network, we evaluated its accuracy and stability using bootstrap testing[32] with data splitting inference[33]. This involves three steps: (1) assessing the accuracy (i.e., 95% confidence interval (CI)) of the estimated coefficients that determine the edges in the network via bootstrap testing, (2) investigating the stability of centrality indices (e.g., in-strength, out-strength, and betweenness) for individual symptoms using case-dropping bootstrap, and (3) testing whether the estimates of coefficients and centralities for different symptoms differ from each other.

**Step 1: assessing the accuracy of coefficients**. To evaluate the accuracy of the estimated coefficients, we can estimate their CIs based on their empirical distributions derived by bootstrap testing. Following bootstrap, a $1 - \alpha$ CI can be approximated by taking the interval between quantiles $1/2\alpha$ and $1 - 1/2\alpha$ of the bootstrapped values. As suggested by Epskamp et al.[32], the type 6 calculation of quantiles is utilized in getting CIs to prevent inflating the type I error. Different from the previous work, in the constructed personal temporal symptom network, the CIs will be drawn for the major effects $\hat{\mu}_j$ and $\hat{\sigma}_{jj'}^1$, as well as for the coefficients of the covariates, including $\hat{\alpha}_{jl}$ and $\hat{\gamma}_{jj'l}^1$.

**Step 2: investigating the stability of centrality indices**. As is discussed in Epskamp et al.[32], bootstrap testing will result in biased empirical distributions for the calculated centralities of a certain network. Therefore, investigation of the stability of centrality indices is proposed to be performed with respect to the order of centralities based on subsets of the data, which is achieved by case-dropping bootstrap. In this paper, we follow this idea to conduct case-dropping bootstrap, in which a proportion of subjects will be randomly dropped for network construction and the centralities will be calculated based on the network estimated using this subset of data. Correlations between the original centrality indices and these obtained from case-dropping bootstrap will be calculated to indicate the stability of the network. The underlying rational is that if the correlation retains high even after dropping a large proportion of data, there is a higher degree of confidence in the interpretation of centrality indices, indicating that the network structure and centrality indices are likely to be reliable.

In contrast to the network estimated in Epskamp et al.[32], the personal temporal symptom network constructed in this study allow for the estimation of a distinct network structure for each subject. This will result in different centrality indices for each individual based on their characteristics, making the calculation of correlation between the original centralities and those obtained from case-dropping bootstrap a very time-consuming task due to the potential large number of subjects in the dataset. To address this challenge, we propose to reduce the total number of individual networks considered for centrality correlation calculation. In practical applications, the covariates $X$ are often represented as categorical or numeric variables. We propose to create representative pseudo-subjects by calculating the mean value of the numeric covariates and generating a unique combination of categorical values for each pseudo-subject. For example, if we have $X = (X_1, \ldots, X_5)$, where $X_1$ is a numeric variable and $X_2, \ldots, X_5$ are binary variables (i.e., level of 0 or 1). Then there will be a total of $2^4 \times 1 = 16$ pseudo-subjects created, each with $X_1$ taking the value of $\bar{X}_1$ and $(X_2, \ldots, X_5)$ taking one of the 16 combinations of $\{0, 1\}^4$ values. These pseudo-subjects can facilitate the identification of network structures that

are representative of the study population. This makes the calculation of the centrality correlations a more feasible task.

**Step 3: testing the differences of coefficients and centralities**. Once the accuracy of the estimated coefficients and the stability of the centrality indices have been evaluated, researchers can perform additional tests to determine whether a specific coefficient is significantly different from another coefficient, or whether the centrality of one node is significantly different from that of another node. These tests can provide further insight into the structure of the symptom network and the relative importance of different symptoms within the network. This can be done by taking the difference between bootstrap values of one coefficient (or centrality) and another coefficient (or centrality) and construct a bootstrapped CI around these difference values. Then a null-hypothesis test can be performed to see whether a certain coefficient or centrality differ from one-another by checking whether zero falls into the bootstrapped CI.

### Diagnostic tests for model assumption evaluation

An essential aspect of inference for the fitted autoregressive logistic regressions with covariates in estimating personal temporal symptom networks is assessing potential violations of statistical assumptions. Various diagnostic tests can be conducted to assess these assumptions, including independence of errors/observations, absence of multicollinearity, fixed predictors, outliers, correctly specified models, and correct functional forms. Additionally, to further evaluate the goodness-of-fit of the fitted models, pseudo R-squared values can be calculated. Details of the diagnostic tests, their implementation, and the interpretation of results on our real data application are provided in Supplementary Note I and Supplementary Tables 1-5.

### Statistics and reproducibility

All statistical analyses were performed using R (version 4.3.3). The logistic autoregressive model with covariates was fitted with the glmnet function from the glmnet package (version 4.1.8). Reported point estimates represent sample averages, and 95% CIs were derived empirically. Details on the sample sizes of pseudo-data used in the simulation studies are provided in the section of **Part 1: Simulation experiments**. For the real data analysis, we used a random sample of 2000 adult survivors of childhood cancer from the SJLIFE cohort. Statistical tests employed to assess model assumptions are described in Supplementary Note I. A significance level of $p < 0.05$ was used for all hypothesis testing.

### Data source for real data analysis

This study utilized secondary data from a random sample of 2000 adult survivors of childhood cancer enrolled in the SJLIFE, a retrospectively constructed cohort designed to prospectively assess adverse health outcomes in survivors[23,24]. Eligible participants were treated for childhood cancer at St. Jude Children's Research Hospital (SJCRH) between 1962 and 2012, survived at least five years post-diagnosis, and were 18 years or older at the time of study participation. Informed consent was obtained from all participants, and the study protocol was reviewed and approved by SJCRH's Institutional Review Board.

### Reporting summary

Further information on research design is available in the Nature Portfolio Reporting Summary linked to this article.

## Results
### Part 1: Simulation experiments

We conducted comprehensive simulation experiments to evaluate the performance of the logistic autoregressive model with covariates in estimating personal temporal symptom networks. The primary simulation results are presented below, with supportive findings provided in Supplementary Note II and Supplementary Table 6.

### Simulation I: identification of associations between symptoms

The first simulation aimed to evaluate the logistic autoregressive model in accurately identifying edges within the temporal network. We simulated networks with 10 nodes, $Y = (Y_1, \ldots, Y_{10}) \in \{0, 1\}^{10}$. Assuming each individual's symptoms were observed at two or more timepoints, the structure comprised of two distinct networks, each with 10 nodes. One network was *static*, capturing associations between nodes at timepoint $t = 1$. Another network was temporal, illustrating associations between nodes at successive timepoints. The adjacency matrix $\Sigma^0$ for the static network was created using the Watts-Strogatz model[32,34] with the number of neighbors set to 2 and the rewiring probability $\theta = 0.3$, producing a small-world network that closely resembled those observed in real-world scenarios. Using this adjacency matrix $\Sigma^0$, where all associations were set to 1, we generated observations at timepoint $t = 1$. Symptom observations at subsequent timepoints ($t \geq 2$) were generated using the temporal network with adjacency matrix $\Sigma^1$. In this matrix, 20 edges were randomly selected to be nonzero. Of these, 15 were set to 1, while the strength of the remaining 5 were determined by individual characteristics, resulting in a personal temporal network. To achieve this, we generated five covariates $X = (X_1, \ldots, X_5)$, where $X_l \sim \text{Bernoulli}(0.5), l = 1 \ldots, 5$. Individualized edges in $\Sigma^1$ were set as $\gamma^1_{jj'l} = \beta X_l, l = 1, \ldots, 5, \beta = 2$. For both the static and temporal network, the thresholds $\mu_j$ were set to $-2$ and for the temporal network, $\alpha_l$ was set to 0. Using the logistic autoregressive model, observations of symptoms were analyzed at timepoints $t = 2, 3$ encompassing a total of 3 timepoints in this simulation. We explored the influence of different sample sizes, setting $n = \{200, 500, 800, 1000\}$. The tuning parameter in eBIC varied as $\eta = \{0.25, 0.5, 1.0\}$. The proposed method was employed to estimate personal temporal symptom networks using simulated datasets. Summary statistics of the edge identification results were derived from 200 replicates. The simulation results, summarized in Fig. 1, present the true positive rate (TPR), false discovery rate (FDR), and Matthew's correlation coefficient (MCC)[35] for detecting all pairwise associations between symptoms in the temporal network, including both covariate-independent and covariate-dependent associations. The findings indicate that TPR and MCC were consistently high, exhibiting a noticeable increase with sample size up to $n = 500$, beyond which the values remained stable. The FPR remained consistently low across different sample sizes. A notable finding is that higher values of $\eta$ led to a more conservative TPR, while MCC exhibited the opposite trend. FPRs were comparable across different $\eta$ values at each sample size.

### Simulation II: evaluation of the stability of centralities

In the second simulation, we evaluated centrality stability by correlating centrality indices (i.e., in-strength, out-strength, and betweenness) obtained from the original dataset's temporal network with those derived from case-dropping bootstrap. In-strength, out-strength, and betweenness centralities quantify the importance of a symptom within a network structure by evaluating its connections to other symptoms. In-strength indicates the sum of edge weights pointing to a node, out-strength indicates the sum of edge weights pointing out from a node, and betweenness counts node occurrences on the shortest path between two other nodes[36]. As we employed data splitting inference for bootstrap, we expanded the sample size to $n = \{400, 1000, 1600, 2000\}$, dividing the sample into two halves, one for network estimation and another for case-dropping bootstrap testing. We simulated the data using the structure of a personal temporal symptom network, with the same settings as in the first simulation. The centralities of nodes in these simulated temporal networks were stable, attributed to variations in edge connections influenced by individual characteristics and node positions determined by the rewiring probability of the Watts-Strogatz model ($\theta = 0.3$). Additionally, we generated unstable temporal networks by removing the effects of individual characteristics on the edges and transforming the temporal network into a ring structure, resulting in nodes having identical centrality values. It is assumed that the correlation of centralities in the stable temporal network would remain high even with a large proportion of case-dropping, while the correlation of centralities in the

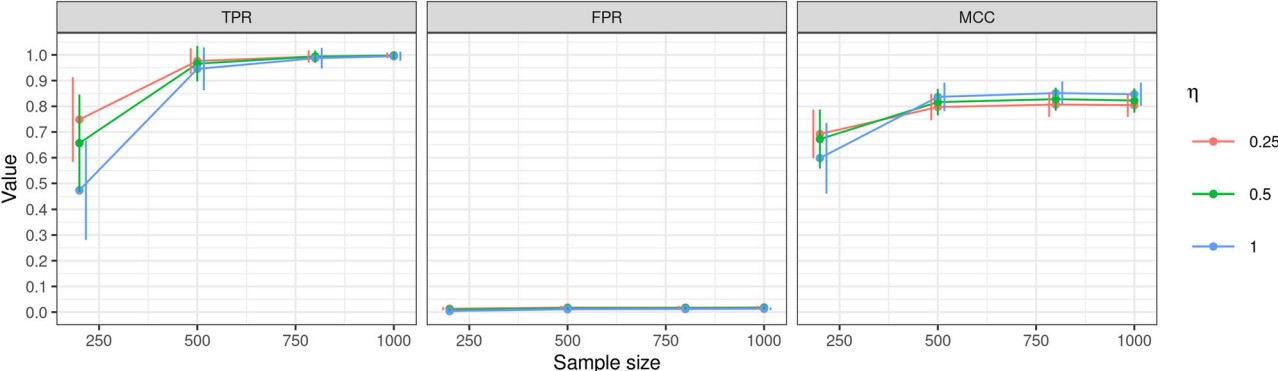

**Fig. 1 | Identification of coefficients in the simulated personal temporal symptom networks.** The evaluation metrics were assessed with different sample sizes. TPR: true positive rate. FPR: false positive rate. MCC: Matthew's correlation coefficient. $\eta$: the tuning parameter of eBIC that controls the network sparsity. The dots represent the average values, and the error bars denote 95% CIs.

unstable temporal network would stay low during the case-dropping bootstrap. All networks were constructed at $\eta = 0.25$. We calculated the Spearman's rank correlations for centralities through 500 bootstraps, varying the sampling proportion as $\Delta = \{0.9, 0.8, 0.7, 0.6, 0.5\}$, and summarized the results based on 200 replicates. Figure 2 illustrates that the correlations of centralities decrease with a larger proportion of case-dropping for the network with stable centralities. However, the correlations for in-strength, out-strength, and betweenness remained high even with a 50% drop in the samples. In contrast, temporal networks with unstable centralities exhibited low correlations (below 0.40), even with a 10% drop in the data, and the trends showed greater variability. This finding implies that the correlation of centralities in case-dropping bootstrap provides a reliable assessment of the stability of centrality indices in an estimated temporal network.

**Simulation III: examination of differences between associations**
In the third simulation, we tested differences between the estimated coefficients and centralities in the constructed unstable temporal networks, using the same settings as described above to generate these unstable networks. Since no edges in the temporal networks were determined by individual characteristics and the networks had a ring structure, no coefficient or centrality should significantly differ from one another. We conducted 500 bootstraps to generate the empirical distribution of the estimated coefficients and centralities and tested the empirical distributions of the differences to check if zero fell within the 95% CI. We calculated the type I error rates for testing these differences across 200 replicates (Fig. 3). The results suggest that the type I error rate consistently decreased with an increase in sample size, and the type I error rate was maintained around 0.05 at $n = 1000$ for both the test of edges and centralities, irrespective of the choice of $\eta$. Additionally, there was a noticeable trend, where decreasing $\eta$ was associated with lower type I error rates across all sample sizes.

**Part 2: Real data analysis**
We applied the logistic autoregressive model with covariates to analyze longitudinal symptom data collected from a random sample of 2000 adult survivors of childhood cancer who participated in the SJLIFE study between 2007 and 2020. This random sample was selected to be consistent with the size of sample used in our previous analysis of baseline symptom data. A detailed introduction to the SJLIFE study can be found in our previous publication[24,37]. Building on our previous work with baseline symptom data[5], this study used symptom data from two timepoints (baseline (T1) and first follow-up (T2)) to enable temporal network estimations. We collected self-reported symptom data from survivors via a 37-item survey capturing ten clinically meaningful domains: cardiac symptoms, pulmonary symptoms, sensation abnormality, nausea, movement problems, pain, memory problems, fatigue, anxiety, and depression. We coded each symptom

domain as present (1) if any symptoms within the corresponding domain were endorsed, and absent (0) otherwise. The logistic autoregressive model incorporated individual heterogeneities as covariates, including socio-demographic characteristics collected at the baseline survey and treatment data abstracted from medical records. We considered six socio-demographic variables: age at the baseline survey (years), sex (male vs. female), race/ethnicity (non-Hispanic White vs. other), attained education (above college/post-graduate vs. below college/post-graduate), annual personal income (≥$20,000 vs. <$20,000), and marital status (married vs. not married), as well as two treatment variables: ever received chemotherapy (no vs. yes) and radiation therapy (no vs. yes). We coded all individual heterogeneities as binary variables, except for the attained age, which was treated as a numeric variable and standardized to a mean of 0 and a standard deviation (SD) of 1. When performing the logistic autoregressive model with covariates, we set the eLasso tuning parameter to $\eta = 0.25$, which yielded good network estimation results based on our simulations. Alternatively, parameter tuning for $\eta$ can be performed using methods such as cross-validation. As in our previous publication[5], we employed data-splitting inference with 100 replicates to summarize the network estimation results.

**Participant characteristics**
The characteristics of survivors randomly selected from the SJLIFE study for personal temporal symptom network estimation are presented in Table 1. The mean age of survivors at T1 was 31.5 years (SD = 8.4). The mean time from diagnosis to T1 was 23.3 years (SD = 8.1), and the mean time between T1 and T2 was 4.4 years (SD = 1.7). Half of the survivors were male (50.2%) and most were non-Hispanic White (84.7%). The majority had education below a college level (62.7%), had personal annual income of <$20,000 (53.9%), and had ever been married or lived as married (62.8%). Leukemia (36.4%) and solid tumors (31.0%) were the most common diagnoses, followed by lymphoma (20.8%) and central nervous system tumors (11.3%). Most survivors had received chemotherapy (86.4%) and/or radiation therapy (60.3%).

Table 2 presents the prevalence of ten symptom domains at T1 and T2. At T1, pain was most prevalent (37.1%), whereas pulmonary symptoms (10.3%) were least prevalent. At T2, the most and least prevalent symptoms were sensation abnormalities (36.9%) and nausea (13.0%), respectively.

**Estimation of associations between symptoms**
We incorporated all individual characteristics and cancer treatment variables from Table 1 into the logistic autoregressive model for estimating the personal temporal symptom network, excluding cancer diagnosis as cancer therapy has a more direct influence on late effects[2,23]. Figure 4 lists the temporal associations between the co-occurrence of pairwise symptom domains over time (i.e., from T1 to T2) in the estimated personal temporal symptom network. The original estimator for the association between

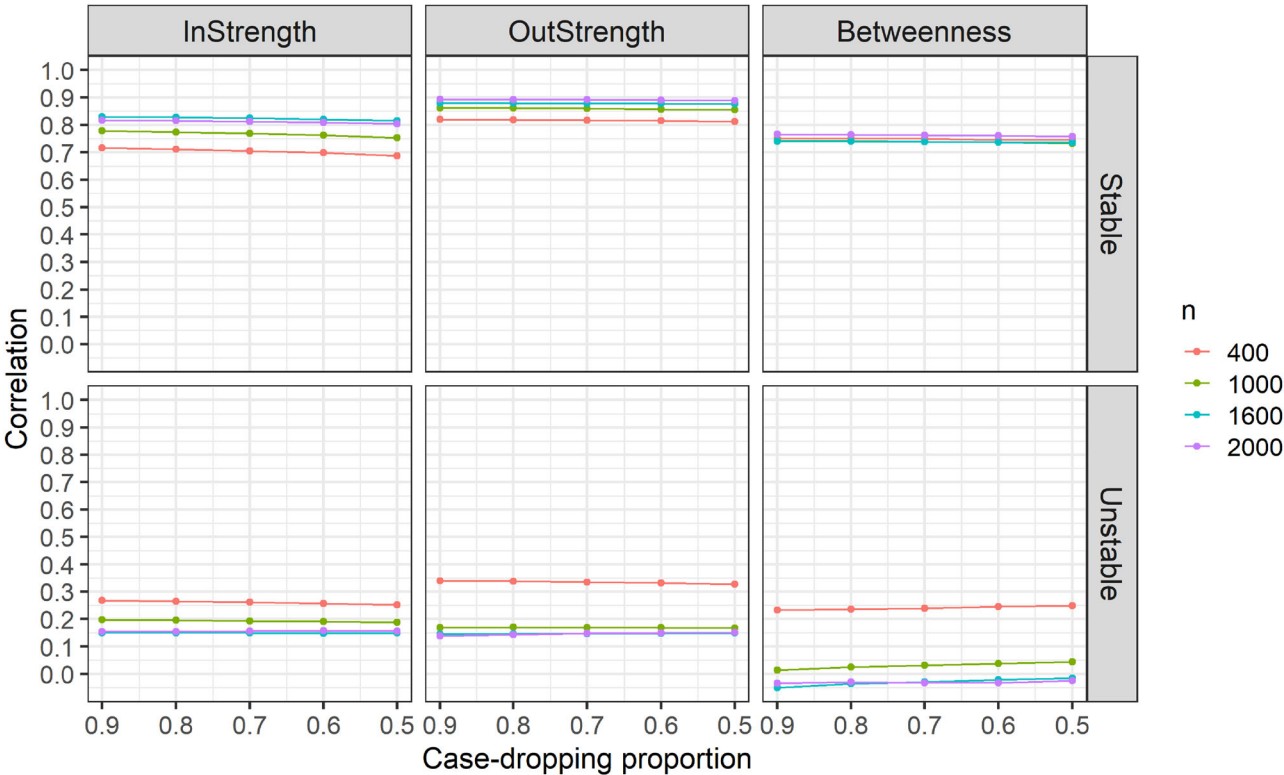

**Fig. 2 | Correlation of centrality metrics (in-strength, out-strength, betweenness) in the simulated personal temporal symptom networks.** The correlations for centrality metrics were evaluated with different case-dropping proportions and sample sizes. The definition of the stable and unstable network structure can be found in the Methods Section – *Step 2: investigating the stability of centrality indices*. The dots represent the average values.

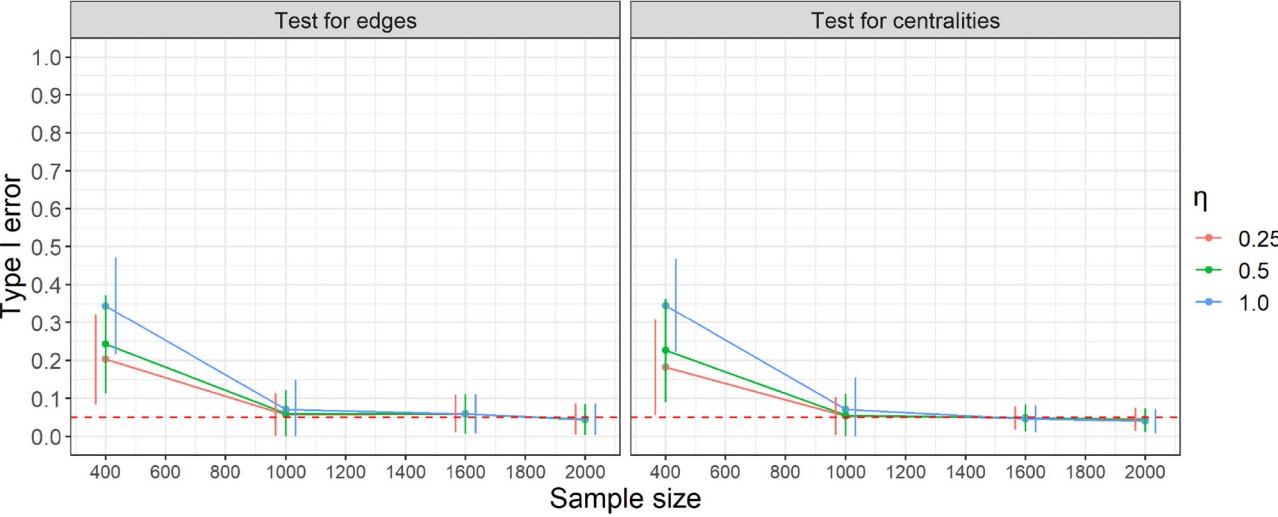

**Fig. 3 | Type I error of testing the differences of edges and centralities for the simulated unstable personal temporal symptom networks.** The type I error was assessed with different sample sizes. $\eta$: the tuning parameter of eBIC that controls the network sparsity. The dots represent the average values, and the error bars denote 95% CIs.

symptoms $Y_j$ and $Y_{j'}$, independent of covariates (i.e., $Y_j \rightarrow Y_{j'}$), represents the log odds ratio (OR) of symptom $Y_{j'}$'s occurrence at T2 when $Y_j$ changes from absence to presence at T1, adjusting for all other symptoms and covariates. The original estimated association between symptoms $Y_j$ and $Y_{j'}$, dependent on covariate $X_l$ (i.e., $Y_j \rightarrow Y_{j'} : X_l$), represents the difference in log OR of symptom $Y_{j'}$'s occurrence at T2 caused by $X_l$ changes from 0 to 1, given that $Y_j$ changes from absence to presence at T1, adjusting for all other symptoms and covariates. In Fig. 4, we report the transformed exponential values – OR and ratio of OR (ROR) – instead of the original log

OR and difference in log OR, to more clearly represent the estimated temporal associations between symptoms over time. OR and ROR values greater than 1 indicate an increased risk of symptom experience.

The strongest associations were identified between symptoms within the same domains over time (i.e., from T1 to T2), independent of the influences of individual heterogeneities. For example, T1 cardiac symptoms were strongly associated with T2 cardiac symptoms (OR: 6.29, 95% CI: 1−14.65), indicating that survivors tend to experience persistent cardiac symptoms over time. It is essential to acknowledge that although the 95% CI

## Table 1 | Characteristics of 2000 adult survivors of childhood cancer included in real data analysis for the personal temporal symptom network analysis

| | | Mean (SD) or N (%) |
|---|---|---|
| Socio-demographic characteristics | | |
| Age at baseline (T1) (years) | | 31.5 (8.4) |
| Time from diagnosis to baseline (T1) (years) | | 23.3 (8.1) |
| Time between baseline (T1) and first follow-up (T2) (years) | | 4.4 (1.7) |
| Sex | Male | 1003 (50.2%) |
| | Female | 997 (49.9%) |
| Race/ethnicity | Non-Hispanic White | 1694 (84.7%) |
| | Other | 306 (15.3%) |
| Attained education | College degree or higher | 747 (37.4%) |
| | Less than college degree | 1253 (62.7%) |
| Annual personal income | ≥$20,000 US dollars | 922 (46.1%) |
| | <$20,000 US dollars | 1078 (53.9%) |
| Marital status | Ever married or lived as married | 1256 (62.8%) |
| | Never married or lived as married | 744 (37.2%) |
| Cancer diagnosis and cancer treatments | | |
| Cancer diagnosis | Leukemia | 728 (36.4%) |
| | Lymphoma | 416 (20.8%) |
| | Central nervous system (CNS) tumor | 225 (11.3%) |
| | Solid tumor | 619 (31.0%) |
| | Histiocytosis | 12 (0.6%) |
| History of chemotherapy | No | 272 (13.6%) |
| | Yes | 1728 (86.4%) |
| History of radiation therapy | No | 795 (39.8%) |
| | Yes | 1205 (60.3%) |

SD standard deviation

## Table 2 | The presence of ten symptom domains at baseline (T1) and the first follow-up (T2) for the 2000 adult survivors of childhood cancer included in the personal temporal symptom network analysis

| Symptom | Baseline (T1) | First follow-up (T2) |
|---|---|---|
| Cardiac symptoms | 302 (15.1%) | 359 (18.0%) |
| Pulmonary symptoms | 205 (10.3%) | 295 (14.8%) |
| Sensation abnormality | 711 (35.6%) | 738 (36.9%) |
| Nausea | 287 (14.4%) | 259 (13.0%) |
| Movement problems | 340 (17.0%) | 403 (20.2%) |
| Pain | 741 (37.1%) | 672 (33.6%) |
| Memory problems | 534 (26.7%) | 598 (29.9%) |
| Fatigue | 365 (18.3%) | 394 (19.7%) |
| Anxiety | 659 (33.0%) | 602 (30.1%) |
| Depression | 604 (30.2%) | 558 (27.9%) |

illustrate the structure of the estimated personal temporal symptom network for two representative survivors: survivor (a) with more favorable risk factors for symptom burden, including male, 31.5-year-old (mean age among all SJLIFE survivors) at T1, a college degree or higher education, an annual personal income ≥$20,000, and no history of treatment with chemotherapy or radiation therapy; survivor (b) with more unfavorable risk factors, including female, 39.9-year-old (one SD above the mean age) at T1, a less than a college degree, an annual personal income <$20,000, and a history of chemotherapy and/or radiation therapy. These two personal networks reveal that survivors with more unfavorable factors exhibit a more complex temporal symptom network than survivors with favorable factors. Compared to survivor (a), the more complex symptom network structure in survivor (b) is evident in its closer interconnections and stronger magnitudes among symptom domains. All the associations determined by individual heterogeneities, as discussed above, are illustrated in Fig. 5b.

### Bootstrap testing
Using two representative survivors displayed in Fig. 5, we calculated the centrality indices (i.e., in-strength, out-strength, betweenness) for each symptom domain in the constructed personal temporal symptom networks and reported the results in Fig. 6. We found that survivor (b), who had more unfavorable risk factors, exhibited higher in-strength, out-strength, and betweenness values for all ten symptom domains, except lower betweenness values for movement problems, pain, and anxiety compared to survivor (a). This finding supports our assertion that survivors with more risk factors possess a temporal network characterized by greater complexity and stronger interconnections between symptom domains over time. Additionally, the high correlations of the centrality rankings obtained from the case-dropping bootstrap demonstrated the stability of the constructed personal temporal symptom network (Fig. 7). The results of bootstrap testing displayed in Supplementary Figs. 1 and 2 further support significant differences in certain temporal associations between symptom domains and in specific centrality measures within the constructed personal temporal symptom network for the survivor (b). For example, the association between T1 and T2 depression differed significantly from the association between T1 fatigue and T2 pain. Although significant differences were identified in some centrality measures (e.g., the out-strength of sensation abnormalities compared to fatigue), the bootstrap testing revealed no significant differences in in-strength and betweenness measures for any of the symptoms.

### Diagnostic tests for model assumptions
Diagnostic tests indicate that the autoregressive logistic regression models with covariates applied in our real data analysis do not exhibit notable

includes one for the associations between some symptom domains, these associations are still considered significant due to using eLasso for variable selection in estimating personal temporal symptom networks[32].

Our analysis further revealed that survivors with annual personal incomes <$20,000 had a higher risk of experiencing T2 memory problems if they had T1 movement problems, compared to those earning ≥$20,000 (ROR: 1.42, 95% CI: 1.0–3.37). We also observed an increased risk of T2 fatigue for lower-income survivors who experienced T1 sensation abnormalities (ROR: 1.35, 95% CI: 1.0–2.97). Older survivors had a higher risk of experiencing T2 cardiac symptoms if they had T1 cardiac symptoms, compared to younger survivors (ROR: 1.10, 95% CI: 1.0–1.86). Female survivors had a higher risk of experiencing T2 nausea if they had T1 sensation abnormalities, compared to male survivors (ROR: 1.13, 95% CI: 1.0–2.40). Survivors with less than a college/post-graduate education had a higher risk of experiencing T2 nausea if they had T1 anxiety, compared to survivors with a college/post-graduate education (ROR: 1.06, 95% CI: 1–2.11). Furthermore, survivors who received any chemotherapy had a higher risk of experiencing T2 pulmonary symptoms if they had T1 pain, compared to survivors who did not receive chemotherapy (ROR: 1.08, 95% CI: 1–2.25). There was an increased risk of T2 pulmonary symptoms for survivors previously exposed to radiation therapy who endorsed T1 pain, compared to those not exposed to radiation therapy (ROR: 1.13, 95% CI: 1–2.45).

### Construction of personal temporal symptom network
We depicted the personal temporal symptom network based on the identified associations between symptom domains. Figure 5a, b

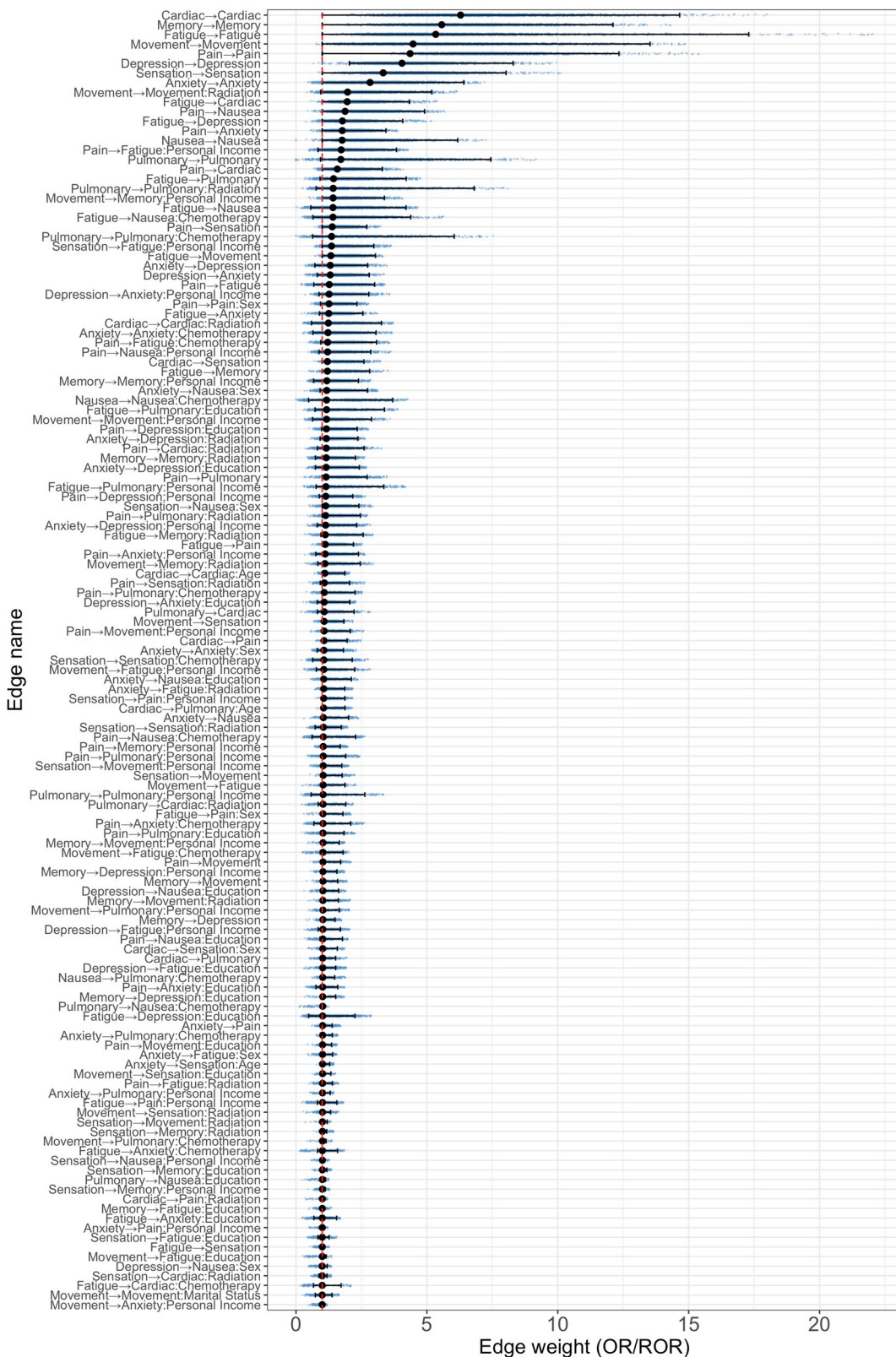

**Fig. 4 | The identified pairwise associations over time between symptoms of the personal temporal symptom network estimated based on the adult survivors of childhood cancer from SJLIFE study ($n = 2000$).** The edge name "$Y_1 \rightarrow Y_2$" represents the identified association between T1 symptoms $Y_1$ and T2 symptom $Y_2$ independent of individual characteristics, and the edge name "$Y_1 \rightarrow Y_2 : X$" represents the identified association between T1 symptom $Y_1$ and T2 symptom $Y_2$ influenced by individual characteristic $X$. The black dots represent the average point estimates, the blue dots represent the point estimates obtained through bootstrap, and the lines represent the corresponding 95% CIs.

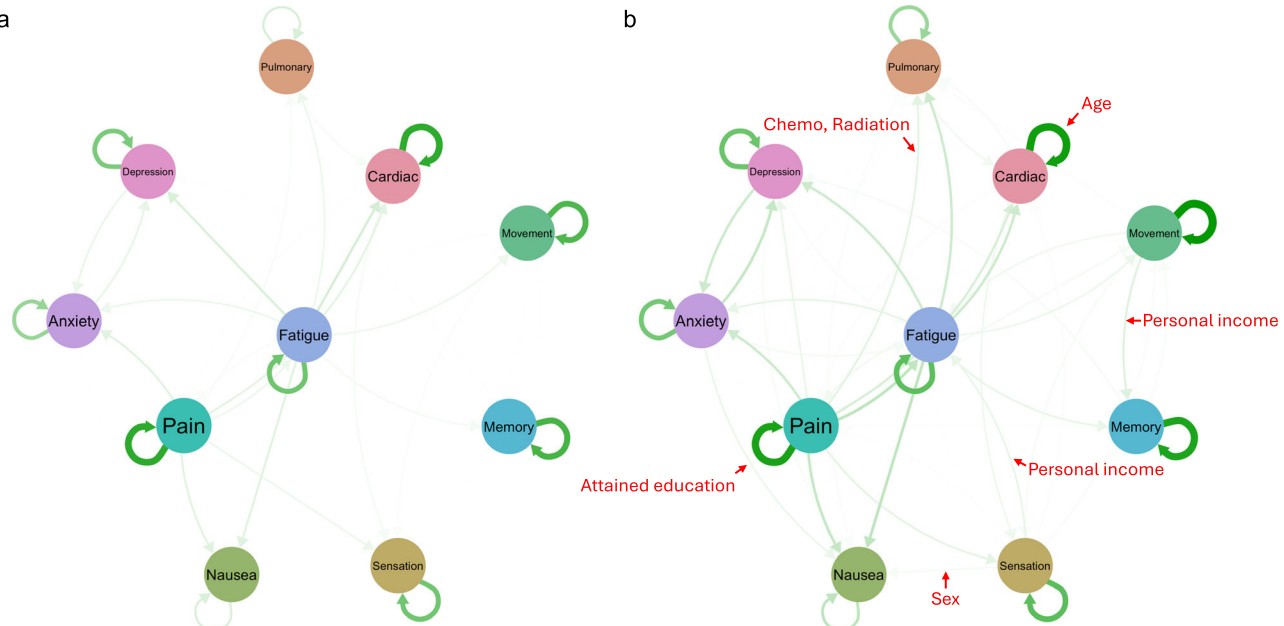

**Fig. 5 | The estimated personal symptom network for two representative cancer survivors. a** A survivor with more favorable risk factors for symptom burden including male, 31.5-year-old (mean age among all SJLIFE survivors) at baseline survey, a college degree or higher, a personal annual income ≥$20,000, and no history of chemotherapy and radiation therapy; survivor. **b** A survivor with more unfavorable risk factors including female, 39.9-year-old (one SD above the mean age) at baseline survey, a less than college degree, an annual income <$20,000, and a history of chemotherapy and/or radiation therapy. Edges pointed to by arrows with covariates indicate the influence of covariates on the pairwise associations between symptoms over time.

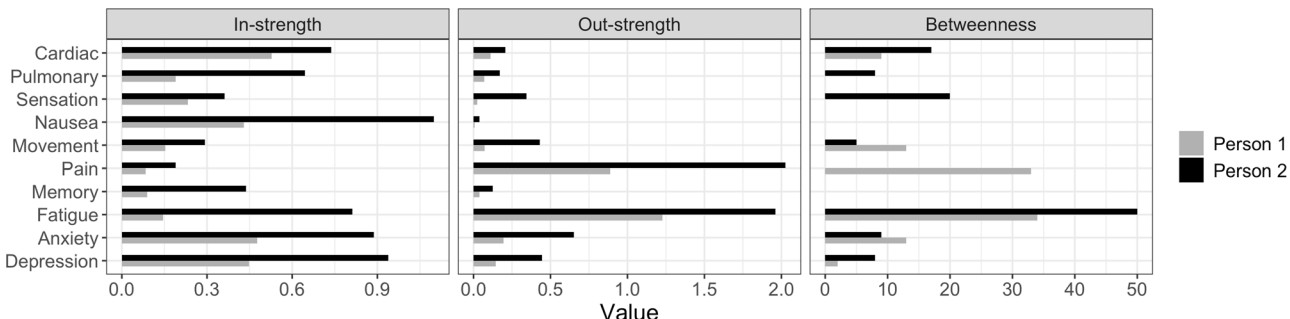

**Fig. 6 | Values of centrality indices (in-strength, out-strength, betweenness) in the personal temporal symptom network for two representative survivors. Person 1:** A survivor with more favorable risk factors for symptom burden including male, 31.5-year-old (mean age among all SJLIFE survivors) at baseline survey, a college degree or higher, a personal annual income ≥$20,000, and no history of chemotherapy and radiation therapy; survivor. **Person 2:** A survivor with more unfavorable risk factors including female, 39.9-year-old (one SD above the mean age) at baseline survey, a less than college degree, an annual income <$20,000, and a history of chemotherapy and/or radiation therapy.

concerns related to autocorrelation, multicollinearity, fixed predictors, or outliers, suggesting that the key model assumptions are satisfactorily met. However, the overall goodness-of-fit assessment indicates a moderate model fit (see Supplementary Note I and Supplementary Tables 1–5).

## Discussion

This study evaluated the performance of the logistic autoregressive model with covariates in constructing personal temporal symptom networks through extensive simulation experiments and its application to estimate the symptom networks for adult survivors of childhood cancer using real data. Simulation results demonstrated that this model can accurately estimate the structure of temporal symptom networks, revealing associations between concurrent symptom domains both with and without the influences of individual characteristics. Real data analysis indicated that individual symptoms tend to persist and correlate with the progression of other symptoms over time. Additionally, individual heterogeneities significantly influenced the onset of symptoms between baseline and follow-up.

As far as we are aware, this study is the first to explore the temporal associations among symptom domains by using personal network structure analysis. It confirms that multiple symptoms experienced by adult childhood cancer survivors over time are interconnected through a complex network system rather than existing independently. Within this network, the status of a symptom at future timepoints may change based on its previous status or the status of neighboring symptoms, highlighting the dynamic and interconnected nature of symptom progression. The discovery of a personal temporal symptom network advances symptom research and clinical management. First, tracking symptom networks longitudinally enables the development of tailored interventions to address health conditions underlying particular symptoms. This proactive approach prevents the worsening of existing symptoms and the emergence of new ones, as indicated by arrows pointing to the target in the symptom network. By identifying the central symptoms within symptom networks, we can design targeted interventions aimed at managing specific late effects, leading to more effective disease management[10]. Second, this approach enhances the management of risk factors for individual survivors (e.g., individual socio-

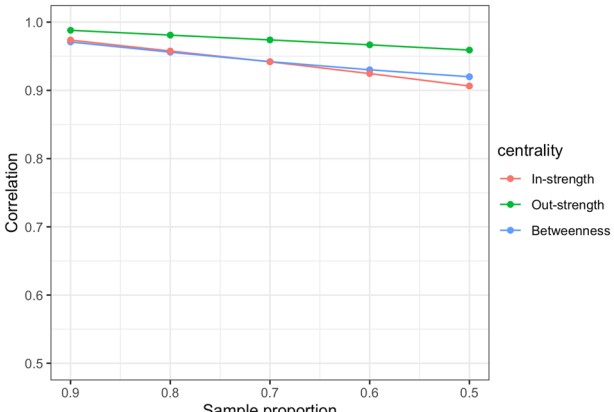

**Fig. 7 | Correlations of centrality metrics (in-strength, out-strength, betweenness) for the personal temporal symptom network estimated based on the adult survivors of childhood cancer from SJLIFE study ($n = 2000$).** The correlations for centrality metrics were evaluated with different case-dropping proportions.

demographic and clinical factors) that significantly strengthen the associations between the target symptom and its neighboring symptoms. For example, our findings reveal a strong association between T1 anxiety and T2 nausea, particularly among survivors with lower education levels (see Figs. 4 and 5). By addressing T1 anxiety in these individuals, clinicians can effectively mitigate the risk of future nausea onset. This targeted strategy, which may include lifestyle modifications and/or medication interventions, is crucial for improving outcomes in vulnerable populations of childhood cancer survivors.

As is mentioned in the *Methods* section, the betweenness metric indicates how often a node appears on the shortest path between two other nodes. In more complex networks, symptoms may have lower betweenness values because the increased number of connections provides more alternative paths, reducing the likelihood of a specific node being on the shortest path. As an example, in a temporal symptom network, if node C lies on the shortest path between nodes A and B, it suggests a temporal predictive relationship from A to C to B. This temporal relationship may not be readily apparent if the logistic autoregressive model is based on data from only two timepoints to predict the current status. However, the current approach shown in the simulation studies provides a way to forecast temporal relationships between multiple nodes if additional observations beyond the second timepoint are available for the construction of the temporal symptom network. This provides a feasible way to understand long-term symptom network trajectories for childhood cancer survivors and inform their symptom management strategies.

Despite the valuable insights provided by our study, several limitations should be acknowledged. First, in the real data application, we only included a small number of adult cancer survivors from the SJLIFE study for personal temporal symptom network estimation. While the sample size of 2000 may seem small, our simulation results (Figs. 1–3) demonstrate that the size of 2000 is sufficient to achieve high accuracy in parameter estimation and edge detection for personal temporal symptom networks, reliable correlation estimates between centrality orders, and adequate type I error control and statistical power for hypothesis testing of edges and centralities. Therefore, despite the sample size constraint, our real data analysis provides a robust estimate of personal temporal symptom networks across two timepoints. Nevertheless, future research, leveraging a larger sample size of adult cancer survivors with longitudinal symptom data collected over multiple timepoints, is needed to further enhance network estimation accuracy. Second, the simulation experiments were conducted for pseudo-data with observations over three timepoints, yet the real analysis of SJLIFE study survivors was limited to two assessments (T1 and T2). This limitation was due to the small number of survivors in the SJLIFE study who had three or more

complete symptom observations (approximately 1200). Based on our simulation results presented in Fig. 3, a sample size of approximately 2000 is required to ensure adequate type I error control and statistical power for hypothesis testing of edges and centralities when performing data splitting inference for data over three timepoints. To improve the robust estimate of personal temporal symptom network across multiple timepoints, future research should apply our approach to datasets with at least 2000 participants who have symptom data spanning three or more assessments. Third, the limitations of using logistic regression to estimate correlations between symptoms are evident. Our symptom network analysis only includes symptoms assessed with binary categories, capturing their presence rather than severity or interference with daily activities, which are often measured on a continuous or ordinal scale. Future research is warranted to extend the current methodology to establish personal symptom networks by accommodating continuous or ordinal data. Meanwhile, logistic regression has inherent constraints in capturing complex, nonlinear relationships between symptoms and is limited in accommodating a large number of risk factors, as the number of parameters increases exponentially with the increase of covariates. Therefore, logistic regression may not be suitable for personal temporal network estimation in case of intricate symptom correlations or when the dimension of covariates and symptoms is larger. As the symptom burden is triggered by different factors at multiple levels (i.e., clinical, personal, family, and neighborhood-level covariates), robust methods capable of handling intricate symptom correlations and a broader range of covariates are needed for analyzing high-dimensional data for personal symptom networks. Exploring machine learning or deep learning algorithms for network estimation presents a promising research direction, especially accommodating intricate symptom correlations and high-dimensional individual risk factors[38].

We also demonstrated that the autoregressive logistic regression models with covariates applied in real data analysis did not violate model assumptions related to autocorrelation, multicollinearity, fixed predictors, or outliers. However, the overall goodness-of-fit assessment suggests only moderate model fit, which may be influenced by various factors, such as the complexity of symptom dynamics, unmeasured confounders, or model specification limitations. While these limitations highlight areas for further refinement, they do not undermine the exploratory value of our approach. Despite the moderate fit, the model successfully identified meaningful symptom network structures and key individual characteristics influencing these networks. Future research is needed to improve this current research by incorporating additional symptom domains to reduce omitted variable bias, extending the autoregressive structure to include longer lag terms, exploring more flexible modeling approaches, such as generalized additive models or machine learning algorithms, to capture nonlinear dependencies, and expanding individual- and neighborhood-level covariates to enhance predictive accuracy and robustness.

## Data availability

The deidentified data are publicly available on Zenodo.org (https://zenodo.org/records/16651889)[39]. These source data for Figs. 4–7 and Supplementary Figs. 1–2 are also provided in the Supplementary Data.

## Code availability

Codes that implement the logistic autoregressive model with covariates are covered by the MIT license and are available on GitHub (https://github.com/SamiraDesh/IndivNA.git)[40]. Tutorials on usage are also provided.

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

## Acknowledgements

The research reported in this manuscript was supported by the U.S. National Cancer Institute under award numbers U01CA195547 (Hudson/Ness), R01CA238368 (Huang/Baker), and R21CA202210 (Huang/Krull).

## Author Contributions

Y.Z. developed the methods, wrote the code, designed the simulations, performed the real data analysis, and wrote the paper; S.D. conducted simulation experiments and maintained the R package on GitHub; M.R.H. and J.C. preprocessed the SJLIFE data; D.A.M., K.K.N., M.M.H., and D.K.S. critically reviewed the results and the manuscript draft; I.C.H. conceptualized the study, interpreted the results, and wrote the paper.

## Competing Interests

The authors declare no competing interests.
