## [Transparent Peer Review file · Communications Medicine]

Estimating Personal Temporal Symptom Networks for Childhood Cancer Survivors

Corresponding Author: Dr Yiwang Zhou

Version 0:

Reviewer comments:

Reviewer #1

(Remarks to the Author)

An important part of the inference is examining the model for indications that statistical assumptions have been violated. This diagnostic process involves a considerable amount of judgement call, because there are not typically statistical tests that can be used to provide assurance that the model meets assumptions or not. The sample data may be expected to depart from what is expected by the assumption even if there is no real violation in the population. Assumptions of the model are interrelated. I would like the authors to address the assessment of the assumptions— independence of errors/observations, correctly specified model (all relevant predictors included), correct functional form, absence of multicollinearity, fixed predictors. Can the methods identify an outlier? Also, is it possible to provide a pseudo r-squared?

Reviewer #2

(Remarks to the Author)

Review

Estimation of Personal Symptom Networks Using the Ising Model for Adult Survivors of Childhood Cancer: A Simulation Study with Real-World Data Application

Zhou, Y., Horan, M. R., Deshpande, S., Ness, K. K., Hudson, M. M., Huang, I.-C., & Srivastava, D.

The authors focused on identifying interconnected symptoms in individual adult survivors of childhood cancer patients by evaluating personal symptom network estimation using the Ising model with covariates. They used Simulation Simulation experiments and real-world data (1,708 adult childhood cancer survivors) to demonstrate the robustness of their model.

The authors propose and evaluate using the Ising model with covariates to construct "personal symptom networks" that account for individual heterogeneities. Through simulations and real-world data (1,708 patients) analysis of the St. Jude Lifetime Cohort Study, the study demonstrates the robustness of the Ising model. It identifies significant individual characteristics (age, sex, race/ethnicity, education, marital status, and treatment history) that influence symptom co-occurrence patterns. The findings provide insights into interconnected symptom experiences and can inform precision-based interventions, i.e., personalized ones.

- Need for Individualized Symptom Analysis: Traditional symptom network analyses often average across individuals, neglecting the significant heterogeneities in symptom experiences. The authors highlight the importance of understanding how individual characteristics influence associations between multiple symptoms.
- Ising Model with Covariates for Personal Symptom Networks: The study utilizes the Ising model with covariates to estimate personal symptom networks. The model allows for incorporating individual characteristics (covariates) into estimating associations between binary symptoms, thereby creating individualized networks. The personal symptom network aims to show "the impact of individual characteristics on the associations between symptoms within a network. This elucidation will provide a distinct structure of a symptom network determined by individual covariates.
- Robustness of the Ising Model: Simulation experiments demonstrate the accuracy and stability of the Ising model with covariates in constructing personal symptom networks. The simulations assess the model's ability to identify true edges, maintain centrality accuracy, and control type I errors.

- Significant Individual Characteristics Influencing Symptom Networks: Real-world data analysis reveals that age, sex, race/ethnicity, education, marital status, and treatment history (chemotherapy and radiation) significantly influence symptom associations in adult survivors of childhood cancer. For instance, "Older childhood cancer survivors showed stronger cardiac-fatigue associations. Survivors of racial/ethnic minorities had stronger pain-fatigue associations. Female survivors with above-college education demonstrated stronger pain-anxiety associations. Unmarried survivors who received radiation had a stronger association between movement and memory problems."
- Implications for Precision-Based Interventions: The estimated personal symptom networks can identify central symptoms and individual characteristics that trigger overall network structure, ultimately informing the design of personalized interventions to improve symptom management.

In general, I find the value of this research very limited, given the following concerns:

1. The model is based on a static network. However, as the authors point out, in cancer, as in many other diseases, "...the symptom burden of individuals with cancer may change over time." In directed or non-directed static networks, if A is directly connected to B and B is directly connected to C, then A is indirectly connected to C. In temporal networks, however, nothing can propagate from A via B to C. Thus, the time ordering can matter greatly. Ignoring the temporal aspect of the network raises serious doubt about the credibility of the work and the validity of the results.
2. The small number of patients (1,708) was used to demonstrate the robustness of the model. This, again, raises doubt about the validity of the results.
3. It is unclear to me how the results based on the simulation and RWD demonstrate the model's robustness.
4. The authors show the confidence intervals; however, they fail to demonstrate how the confidence intervals illustrate the accuracy of the networks.
5. The lack of medical justification for the symptom associations identified by the model.

Version 1:

Reviewer comments:

Reviewer #2

(Remarks to the Author)

I am satisfied with the authors' response to my comments and I recommended the publication of the revised version of the paper.

Reviewer #3

(Remarks to the Author)

The authors have engaged in the process and they have addressed the comments appropriately. No further comments.

Point-to-Point Responses to Referee 1

We would like to express our gratitude for your review and careful reading of our paper. Below are listed point-to-point responses to your comments presented in italics.

Referee's Comment: An important part of the inference is examining the model for indications that statistical assumptions have been violated. This diagnostic process involves a considerable amount of judgement call, because there are not typically statistical tests that can be used to provide assurance that the model meets assumptions or not. The sample data may be expected to depart from what is expected by the assumption even if there is no real violation in the population.

Assumptions of the model are interrelated. I would like the authors to address the assessment of the assumptions—independence of errors/observations, correctly specified model (all relevant predictors included), correct functional form, absence of multicollinearity, fixed predictors. Can the methods identify an outlier?

Also, is it possible to provide a pseudo r-squared?

Our Response: We appreciate your suggestion to evaluate the model assumptions. Accordingly, for the real-data application, we have conducted the following diagnostic tests to assess these assumptions.

1. **Independence of errors/observations:** Since the autoregressive logistic model estimates correlations between symptoms observed at the previous and current timepoints, it is essential to assess whether the residuals from the time series model exhibit autocorrelation (i.e., whether they are independent over time). To evaluate this, we applied the Ljung-Box test, where the null hypothesis states that the residuals are independently distributed (i.e., no autocorrelation), and the alternative hypothesis indicates the presence of autocorrelation. In our real-data application, we conducted 100 iterations of data splitting inference. For each iteration, we performed the Ljung-Box test on the residuals for each of the ten symptom domains and computed the mean p-value across all iterations. The summarized p-values, presented in Supplementary Table 2, indicate that the null hypothesis was not rejected, suggesting no evidence of autocorrelation in the residuals.

Supplementary Table 2. The mean p-values obtained from the Ljung-Box tests.

Symptom	p-value
Cardiac	0.408
Pulmonary	0.441
Sensation	0.517
Nausea	0.478
Movement	0.510
Pain	0.381
Memory	0.229
Fatigue	0.511
Anxiety	0.518
Depression	0.523

- Absence of multicollinearity:** Assessing the absence of multicollinearity in autoregressive logistic regression models with LASSO can be challenging because LASSO inherently performs variable selection and shrinks correlated predictors toward zero, thereby reducing concerns about multicollinearity. However, some degree of multicollinearity may still exist among the selected predictors. To evaluate this, we calculated the Variance Inflation Factor (VIF) for the variables retained by LASSO in the fitted autoregressive logistic regression models. VIF quantifies the extent to which the variance of a regression coefficient is inflated due to multicollinearity among predictor variables, helping to assess whether independent variables are highly correlated. Similar to the Ljung-Box test, we computed the mean VIF for the selected variables in the regression models of each of the ten symptom domains across 100 iterations of data-splitting inference. The averaged VIF values, presented in Supplementary Table 3, are all below 5, suggesting that multicollinearity is not a concern in our autoregressive logistic regression models.

Supplementary Table 3. The average VIF values.

Symptom	VIF
Cardiac	1.867
Pulmonary	2.823
Sensation	2.221
Nausea	2.550
Movement	2.438
Pain	1.585
Memory	2.116
Fatigue	2.442
Anxiety	2.752
Depression	2.394

- Fixed predictors:** In our autoregressive logistic regression model, fixed predictors are those that remain constant over time. These include individual characteristics (i.e., age at the baseline survey, sex, race/ethnicity, attained education, annual personal income, marital status, and ever receiving chemotherapy and radiation therapy). Assessing fixed predictors in an autoregressive logistic regression model with LASSO typically involves examining their consistency in selection across different models and evaluating their impact on the outcome. Since we performed 100 iterations of data-splitting inference and used bootstrapping to construct 95% confidence intervals (CIs) for the estimated coefficients, our data analysis procedure inherently assessed the role of fixed predictors in the fitted models. The identified individual characteristics that significantly influence the correlations between symptoms highlight the importance of fixed predictors in understanding symptom dynamics over time.
- Outlier:** Identifying outliers in an autoregressive logistic regression model with LASSO requires examining influential observations that may disproportionately affect model estimates. In our real-data application, we employed standardized residuals to detect potential outliers in the fitted autoregressive logistic regression model. Specifically, we computed the standardized deviance residuals and identified observations with absolute values greater than 3, as these may indicate potential outliers. Consistent with the Ljung-

Box test performed above, we calculated the residuals for fitted models across each of the ten symptom domains over 100 iterations of data-splitting inference. The average number of observations with absolute standardized residuals greater than 3, as presented in Supplementary Table 4, is zero or nearly zero, suggesting that outliers are not a concern in our analysis.

Supplementary Table 4. The average number of outliers.

Symptom	Number of Outliers
Cardiac	0.00
Pulmonary	0.00
Sensation	0.00
Nausea	1.11
Movement	0.00
Pain	0.00
Memory	0.00
Fatigue	0.00
Anxiety	0.00
Depression	0.00

- 5. Correctly specified models and correct functional forms:** To evaluate whether the autoregressive logistic model is correctly specified and has the appropriate functional form, we performed the Hosmer-Lemeshow (HL) goodness-of-fit test, which assesses how well the predicted probabilities align with observed outcomes by dividing the data into deciles of risk and comparing expected versus observed event rates. The null hypothesis states that the model is correctly specified, meaning there is no significant difference between observed and predicted probabilities, while the alternative hypothesis suggests model misspecification due to a mismatch between observed and predicted values. Similar to the Ljung-Box test, we applied the HL test to each of the ten symptom domains and computed the mean p-value across 100 iterations of data-splitting inference. The summarized p-values, presented in Supplementary Table 5, indicate that the null hypothesis was rejected, suggesting evidence of model misspecification. This result is not surprising, given the strong assumptions made in modeling symptom correlations over time using the autoregressive logistic model. Several factors may contribute to model misspecification. First, the model only includes a limited set of symptom domains for temporal network construction, and the omission of other relevant symptoms may lead to misspecification. Second, the model assumes that symptoms at the current timepoint depend only on symptoms at the immediately preceding timepoint (lag-1 factorization), whereas symptoms from two or more prior timepoints may also influence current symptoms, a limitation of the current dataset. Third, logistic regression assumes a linear relationship between predictors and the log-odds of the outcome, which is a strong assumption that may not hold in reality; nonlinear relationships may better capture symptom interactions, and alternative approaches such as generalized additive models or machine learning algorithms may provide a more flexible framework. Fourth, due to the model’s complexity, we included only a few individual characteristics such as basic demographics and socioeconomic status, while a more comprehensive set of patient-specific factors plus contextual factors may improve model specification. Therefore, future evaluation is warranted by incorporating additional symptom domains to reduce omitted variable bias, extending the autoregressive structure to include longer lag terms

or more timepoints, exploring more flexible modeling approaches such as generalized additive models or machine learning algorithms to capture nonlinear dependencies, and expanding individual, family, and neighborhood-level covariates to enhance predictive accuracy and robustness. While there is evidence of model misspecification, this study is the first to explore the estimation of personal temporal symptom networks using an autoregressive logistic framework. Our primary goal is not to determine whether the autoregressive logistic model is perfectly specified, but rather to expand this field by gaining insight into how symptoms evolve and correlate over time. Despite this limitation, our findings provide valuable insight into understanding temporal symptom dynamics and lay the foundation for future methodological advancements in this area.

Supplementary Table 5. The mean p-values obtained from the HL tests.

Symptom	p-value
Cardiac	1.41×10^{-3}
Pulmonary	3.90×10^{-4}
Sensation	3.56×10^{-3}
Nausea	1.08×10^{-3}
Movement	3.59×10^{-5}
Pain	9.78×10^{-4}
Memory	3.04×10^{-3}
Fatigue	3.33×10^{-4}
Anxiety	6.19×10^{-3}
Depression	5.12×10^{-4}

- 6. Pseudo R-squared:** Pseudo R-squared is another measure of goodness-of-fit for autoregressive logistic regression models with LASSO. Here, we report the results of McFadden’s R-squared. Consistent with the tests performed above, we calculated McFadden’s R-squared for the fitted autoregressive logistic regression models across each of the ten symptom domains over 100 iterations of data-splitting inference. The average McFadden’s R-squared values, presented in Supplementary Table 6, range from 0.1 to 0.2, suggesting a moderate model fit. This finding aligns with the results reported in Table 5 based on the HL goodness-of-fit test. Potential reasons for the lack of model fit and possible approaches to address this issue in future research have been discussed above.

Supplementary Table 6. The average McFadden’s R-squared values.

Symptom	Pseudo R-squared
Cardiac	0.2
Pulmonary	0.1
Sensation	0.1
Nausea	0.1
Movement	0.2
Pain	0.2
Memory	0.2
Fatigue	0.1
Anxiety	0.1
Depression	0.1

To sum up, we acknowledge the importance of evaluating model assumptions. Our diagnostic tests indicate that the autoregressive logistic regression models do not exhibit significant issues related to autocorrelation, multicollinearity, fixed predictors, or outliers. However, the models demonstrate moderate goodness-of-fit, which may be attributed to various factors discussed in detail above. While we acknowledge the model's limitations in fit, our primary objective is not to achieve a perfect model specification, but rather to explore the application of autoregressive logistic regression with covariates for estimating personal symptom networks. Despite imperfections in fit, our analysis has revealed valuable insights into the structure of the symptom network and identified key individual characteristics that influence these networks.

To evaluate these model assumptions, we have added a subsection in the Methods section describing the evaluation methods, a subsection in the Results section summarizing the findings, and detailed results from our real-data application in the Supplementary Materials. In addition, we incorporated these discussion points into the Discussion section, especially suggesting the need for model fit improvement in future research.

Point-to-Point Responses to Referee 2

We appreciate your review and comments that helped us improve our manuscript. Below are listed point-to-point responses to your comments presented in italics.

Referee's Comment: *This research introduces and validates a "logistic autoregressive model with covariates" to estimate "personal temporal symptom networks." This approach moves beyond analyzing individual symptoms in isolation. Instead, it focuses on how symptoms influence each other over time while considering individual patient characteristics (socio-demographic factors, treatment history). The study uses simulations and real-world data from the St. Jude Lifetime Cohort Study (SJLIFE) to demonstrate the model's effectiveness and potential for personalized interventions.*

Strengths.

- ***Symptom Networks:*** *The paper highlights traditional symptom research limitations that focus on individual symptoms. Such an approach cannot directly infer underlying interconnections between multiple symptoms that can be estimated through a network approach.*
- ***Temporal Dynamics.*** *The research emphasizes the importance of understanding how symptom networks evolve over time. The method utilizes time-series data to model how a symptom at one point predicts the presence or absence of other symptoms later.*
- ***Personalization.*** *A key innovation is the incorporation of individual patient characteristics (covariates) into the network model. This allows for the creation of personal symptom networks that reflect each survivor's unique risk factors and experiences.*
- ***Potential for Targeted Interventions.*** *The ultimate goal is to use personal symptom networks to develop more effective and personalized interventions for managing late effects in childhood cancer survivors. By identifying "sentinel symptoms" and understanding how individual characteristics influence symptom relationships, clinicians can target interventions to specific patients and specific symptom combinations.*

Our Response: Thank you for your thoughtful and positive assessment of our work. We greatly appreciate your recognition of the importance of our proposed autoregressive logistic regression model with covariates in estimating personal temporal symptom networks and its potential for enhancing targeted symptom management and interventions.

Referee's Comment: Weakness.

- *The analysis was based on only two time points: the baseline and the first follow-up; this presents a serious limitation.*
- *Although the study uses real-world data, the number the small sample of cancer patients is a serious limitation.*
- *When dealing with complex data, as is the case with cancer patients' data, logistic regression represents a limitation.*

Our Response: We acknowledge that a key limitation of our real-data analysis for estimating personal temporal symptom networks is its reliance on symptom data collected from only two timepoints. As discussed in the Discussion section, this constraint arose because fewer than 1,200 survivors in the SJLIFE study had three or more complete symptom observations. Based

on our simulation results presented in Figure 3, a sample size of approximately 2,000 is required to ensure adequate type I error control and statistical power for hypothesis testing of edges and centralities when performing data splitting inference for the symptom assessment beyond two timepoints. Therefore, to improve the robust estimation of personal temporal symptom network across multiple timepoints, applying our approach to datasets containing at least 2,000 participants with symptom data spanning three or more timepoints in future research is needed.

The second weakness in the real-data analysis pointed out by the reviewer is also related to the sample size of survivors (2,000) with symptom data collected from only two timepoints. While the sample size of 2000 may seem small, our simulation demonstrates that 2,000 is sufficient to achieve high accuracy in parameter estimation and edge detection for personal temporal symptom networks, reliable correlation estimates between centrality orders, and adequate type I error control and statistical power for hypothesis testing of edges and centralities. Despite the sample size constraint, our real-data analysis provides a robust estimate of personal temporal symptom networks across two timepoints. Nevertheless, future research, leveraging a larger sample size of adult cancer survivors with symptom data collected over multiple timepoints, is needed to further enhance the accuracy of network estimation.

We acknowledge that the use of logistic regression to model correlations between symptom domains for estimating personal temporal symptom networks may present an additional limitation. As discussed in the Discussion section, logistic regression has inherent constraints in capturing complex, nonlinear relationships between symptoms and is limited in accommodating a large number of risk factors/covariates, as the number of parameters increases exponentially with the increase of covariates. To address these challenges in high-dimensional data settings, future work is needed to explore more flexible and robust methods, such as machine learning or deep learning algorithms, that can capture intricate symptom interactions and incorporate a broader range of covariates.

We have discussed these valuable points as limitations in the Discussion section.